# Q Fever Vaccine Development: Current Strategies and Future Considerations

**DOI:** 10.3390/pathogens10101223

**Published:** 2021-09-22

**Authors:** Carrie Mae Long

**Affiliations:** Laboratory of Bacteriology, Division of Intramural Research, National Institute of Allergy and Infectious Disease, National Institutes of Health, Hamilton, MT 59828, USA; carrie.long@nih.gov; Tel.: +1-406-375-9816

**Keywords:** *Coxiella burnetii*, bacterial vaccines, vaccine, intracellular pathogen, Q-Vax, hypersensitivity, Q fever

## Abstract

Q fever is a zoonotic disease caused by the intracellular pathogen *Coxiella burnetii*. This disease typically manifests as a self-limiting, febrile illness known as acute Q fever. Due to the aerosol transmissibility, environmental persistence, and infectivity of *C. burnetii*, this pathogen is a notable bioterrorism threat. Despite extensive efforts to develop next-generation human Q fever vaccines, only one vaccine, Q-Vax®, is commercially available. Q-Vax® is a phase I whole-cell vaccine, and its licensed use is limited to Australia, presumably due to the potential for a post-vaccination hypersensitivity response. Pre-clinical Q fever vaccine development is a major area of interest, and diverse approaches have been undertaken to develop an improved Q fever vaccine. Following a brief history of Q fever vaccine development, current approaches will be discussed along with future considerations for an improved Q fever vaccine.

## 1. A Brief History of Q Fever Vaccine Development

Q fever is a zoonotic disease that typically presents as an acute, febrile, self-limiting disease but can also lead to more chronic manifestations such as endocarditis or vasculitis [1]. Q fever is caused by the intracellular bacterium *Coxiella burnetii* and is typically acquired by inhalation of infectious aerosols generated from livestock. Although *C. burnetii* can cause disease in various organisms, this review focuses on Q fever vaccine development in the context of human disease. *C. burnetii* is endemic nearly worldwide [2] and is considered an occupational hazard among laboratory, livestock, and veterinary workers. Due to the bacterium’s environmental resistance, infectivity, and ability to cause debilitating disease, *C. burnetii* is classified as a Select Agent by the United States Centers for Disease Control and Prevention’s Division of Select Agents and Toxins (DSAT) and is a noted bioterrorism pathogen of interest [3]. Accordingly, Q fever vaccine development commenced almost immediately following the identification of *C. burnetii* in Australia and the United States in the late 1930s. Early Q fever vaccine development efforts were led by NIH scientists at Rocky Mountain Laboratories (RML) [4]. After years of vaccine development efforts among US government scientists, a Q fever vaccine was introduced in humans in 1948 [5]. The vaccine was comprised of whole-cell, formaldehyde-inactivated, ether-extracted *C. burnetii* with 10% egg yolk sac [5]. These vaccines were derived from Dyer and Henzerling *C. burnetii* strains. Initial injections were marked by severe adverse reactions, yet following a series of dilutions, the vaccine was generally well tolerated. This vaccine and related iterations likely reduced the risk of *C. burnetii* laboratory-acquired infections since their inception [6]. Additionally, *C. burnetii* whole-cell vaccine (WCV) efficacy in humans was directly tested, demonstrating protection against high-dose aerosol challenge in US Army volunteers [7].

Early egg derived WCVs displayed remarkable efficacy but were also associated with post-vaccination hypersensitivity (PVH) responses characterized by localized induration, sterile abscesses, and granulomas at the site of injection [8,9] along with systemic responses [5]. These adverse responses are thought to occur exclusively in pre-immune individuals [10]. An additional safety concern with early vaccines was the amount of egg protein contained in the vaccine mixture. In response to these concerns, several adaptations to the early WCVs were adopted. Ormsbee et al. developed a protocol that decreased egg protein contamination via sedimentation prior to ether extraction [11]. Potency and purity standards were developed and introduced by Lackman et al., further refining the use of early Q fever vaccines [12,13]. Pre-vaccination skin testing protocols were introduced in the late 1950s [14] and continue to be an important component of Q fever vaccination screening methods. There was renewed interest in Q fever vaccine development in the 1970s, likely due to the increased Q fever incidence and outbreaks in research and medical settings as well as abattoirs that processed feral goats [15]. Building on these refinements, a chloroform–methanol residue (CMR) vaccine was introduced in 1982 [16] following early work involving bacterial fractionation to determine protective and reactive components. Eventually, the CMR vaccine was tested in human clinical trials and used as an investigational new drug (BB-IND-3516) in the US; this vaccine demonstrated reduced reactogenicity and promising immunogenicity after primary vaccination and boost [17,18] but was not pursued for wide-scale production or pharmaceutical approval. 

Full-length or phase I lipopolysaccharide (LPS) was identified as an important protective factor in vaccine-induced immunity [11,19,20]. The concept of *C. burnetii* LPS phase variation was first described in 1956 by Stoker and Fiset [21]. This phenomenon is characterized by alterations in LPS length based on culturing conditions which, in turn, affects *C. burnetii* virulence [22,23] and WCV potency [24]. Historically, LPS truncation occurred following passage in embryonated egg yolks, but this process can also occur following cellular infection and axenic culture [25]. After the identification of phase variation, the Henzerling-based WCV in use was analyzed and found to be primarily expressing phase II LPS [26]. In response, Henzerling vaccine stocks were converted to phase I LPS expression via in vivo passage and used in an updated phase I vaccine [27]. Although, truncated LPS (phase II)-based vaccines have not gained traction, some phase II-based vaccines, such as the live M-44 variant of the Grita strain, have demonstrated immunogenicity in humans [28] and protective efficacy in guinea pigs [29]. Recent studies have bolstered the hypothesis that the phase I LPS *O*-antigen is a crucial component of *C. burnetii* virulence and vaccine-induced protection [24], yet the importance of phase II LPS in these processes should not be underappreciated.

The RML-derived Q58-A WCV was used as an investigational new drug (US BB-IND-26) [30] following on-site clinical trials and was manufactured by the National Drug Company (Philadelphia, PA, USA). Currently, the only licensed vaccine available, Q-Vax® (Seqirus UK Limited, Maidenhead, UK), is an iteration of the RML WCV. This vaccine, which consists of phase I, formalin inactivated Henzerling WCV, was developed in Australia and initially produced by Commonwealth Serum Laboratories (CSL). Q-Vax^®^ has demonstrated nearly 100% efficacy in human trials [31] and was licensed for use in Australia in 1989 for high-risk groups [32]. Importantly, Q-Vax^®^ requires pre-vaccination screening including serology, patient history, and a skin test. The potential for hypersensitivity reactions following vaccination has likely precluded licensing of Q-Vax^®^ beyond Australia. 

Overall, Q-Vax^®^ has demonstrated durable protective efficacy, but its widespread use has been hampered by the potential for severe adverse reactions and the cumbersome pre-screening process associated with vaccine distribution. As the only licensed vaccine available for human Q fever pre-exposure prophylaxis, Q-Vax^®^ was deployed in the 2011 Q fever outbreak in the Netherlands [33]. This program was met with significant challenges but appeared to be effective in preventing potential Q fever spread. Recent interest in the development of an improved Q fever vaccine has been spurred by natural outbreaks, post-2001 bioterrorism concerns, military concerns, and occupational infection risk [34,35]. Modern Q fever vaccine development efforts have harnessed recent advances in Q fever biology including the introduction of an axenic culture media [36], the resultant development of numerous genetic tools [37], and insights into host–pathogen interactions facilitated by animal modeling. Despite these advances, diverse vaccine development approaches have been employed in the pursuit of an improved Q fever vaccine, but none have surmounted the potency of Q-Vax^®^. With a better understanding of *C. burnetii*–host interaction, including identification of immunogenic antigens and correlates of protection, Q fever vaccine development will be greatly enhanced.

## 2. Correlates of Protection and Immunologic Considerations for an Effective Q Fever Vaccine

For effective vaccine design and implementation, identification of protective correlates of immunity is integral. Typically, these correlates involve protective immunologic components of the vaccine response that are required, responsible for, or closely related to protection [38]. Historically, antibody levels and functional immune metrics have served as correlates of protection for pathogens such as the measles and smallpox viruses. Important considerations for the process of defining correlates of vaccine-mediated protection include defining protective ability throughout the biological disease process (e.g., correlates of protection from infection, dissemination, morbidity, and mortality may be distinct) and dose–responsive effects on correlates of protection (e.g., the impact of vaccine and infectious doses). In the case of primary Q fever and *C. burnetii* vaccination, investigating and defining correlates of protection following both primary infection and secondary immunity are worthwhile pursuits. Indeed, these results may vary depending on factors such as the vaccine type, vaccine or challenge dose, and host organism, but a general understanding of WCV-induced immunity and corresponding protective factors are crucial for the design of an improved Q fever vaccine.

### 2.1. Rodent Models

Generally, adaptive immune responses are associated with vaccine-mediated protection, with antibody titers and cellular readouts representing primary correlates of protection for many vaccines. For *C. burnetii*, the adaptive immune response appears to be imperative for protective responses following both vaccination and natural infection. In murine models, an early role for T and B cells was established in primary clearance following infection, with T cells demonstrating importance in pathogen control and elimination [39,40]. The ability of passively transferred immune sera from WCV vaccinated mice to provide convincing protection in pre-immune mice in response to virulent *C. burnetii* challenge further emphasizes the potential protective capability of B cells following *C. burnetii* vaccination [41,42]. Adoptive transfer of immune sera or splenocytes from phase I WCV vaccinated mice into naive mice reduced splenomegaly at a level comparable to that of phase I WCV administration itself prior to challenge [24], further supporting a role for both cell-mediated and humoral immune responses in WCV-induced protective responses. Further, this study demonstrated that only adoptive transfer of immune splenocytes or T cells into SCID mice could prevent splenomegaly and reduce splenic bacterial burden upon *C. burnetii* challenge, something that transfer of neither sera nor B cells could do. Contrastingly, both cellular and humoral components were able to reduce clinical disease, as reflected by a lack of significant weight loss in SCID mice following transfer and challenge. Ultimately, this study demonstrated the overall importance of cell-mediated immunity for host clearance of *C. burnetii* in a murine model contrasted by the dispensability of humoral immunity for this function. Similarly, Read et al. demonstrated that SCID mice reconstituted with CD4^+^ or CD8^+^ T cells were able to control pulmonary infection better than their non-reconstituted counterparts or those reconstituted with B cells [43]. Indeed, humoral immunity seems to play a distinct role in primary clearance and WCV-induced protection, with critical importance in the prevention of clinical disease. The roles of B cells and antibody in this response should be further elucidated, particularly in relation to potential interactions with T cells (e.g., T follicular helper cells).

Together these studies indicate that T-cell responses are critical for bacterial clearance following infectious challenge. Building on these findings, murine models revealed roles for MHC-I and MHC-II molecules in primary *C. burnetii* infection clearance [44] and MHC-II CD4^+^ T-cell-dependent and -independent mechanisms in vaccine-induced protective responses [45]. Further, a role was established for the transcription factor Tbet in a murine pulmonary *C. burnetii* infection model, with *Tbet*^−/−^ mice displaying reduced ability to control bacterial spread, experiencing persistent infection and increased granuloma formation [46]. These findings are supported by that of a murine vaccine–challenge study demonstrating the induction of Th1 responses following phase I and II WCV vaccination and challenge [24]. Despite the dampened protective efficacy of phase II WCV, Th1 responses of comparable magnitude were observed compared to phase I WCV. A recent study demonstrated enhanced interferon-γ (IFN-γ)-producing CD4^+^ T cells from secondary lymphoid organs in mice vaccinated with phase I WCV compared to that of phase II WCV vaccinated mice [47]. Similar findings were reported in a study utilizing murine infection models with various genetically altered strains [48]. This study revealed that IFN-γ is required for early protection against primary *C. burnetii* infection. Tbet has demonstrated significance beyond the orchestration of CD4^+^ Th1 responses [49], and Th1-independent roles for Tbet were demonstrated in a murine *C. burnetii* phase I WCV vaccine–challenge model [45]. Accordingly, *Cd4*^−/−^ and *Tbet*^−/−^ mice demonstrated variable protection following vaccination and challenge, with *Tbet*^−/−^ mice appearing to experience more severe disease and less efficient bacterial clearance. These data suggest that Tbet and Th1-mediated responses are involved in both primary and WCV-induced *C. burnetii* clearance in murine models, although more investigation is warranted to elucidate specific underlying mechanisms. The apparent importance of T-cell-independent responses involving Tbet and MHC-II illustrate the complexity of protective responses. In a guinea pig model, draining lymph node cellularity and T-cell metrics were distinct two weeks following intraperitoneal infection in naive and phase I WCV vaccinated animals, indicating heterogeneity in primary and secondary immune responses [50]. Overall, specific mechanisms of primary and secondary immune-mediated clearance of *C. burnetii* remain to be fully investigated, particularly in relation to T-cell subset behavior and T-cell-independent processes. 

Macrophages are a major target cell for *C. burnetii* infection and have been studied extensively in this context. Antigen-presenting cells, such as macrophages and dendritic cells, are known to contribute to vaccine-mediated immunity to various intracellular pathogens, primarily due to the fact of their role in bridging innate and adaptive immune responses and initiating cell-mediated immune responses. Indeed, cell-mediated immunity in response to *C. burnetii* phase I WCV has been associated with T-cell-dependent activation of macrophages in a guinea pig model [51,52]. Murine bone-marrow-derived dendritic cells (BMDCs) stimulated with recombinant *C. burnetii* protein antigens (i.e., Com1 and Mip) were able to confer slight protective effects (as measured by reduced early splenic *C. burnetii* load in challenged mice) following vaccination [53]. Although this vaccine strategy may not be technically feasible, these results suggested a possible role for dendritic cells in vaccine-mediated protection against *C. burnetii.* In a separate study, BMDCs from C57Bl/6 mice were simulated with phase I or II WCV ex vivo and assayed for maturation markers and migratory ability, with phase I WCV stimulated cells displaying increased maturation and migratory ability [47]; these findings corresponded with larger draining lymph node size in mice given phase I WCV stimulated BDMCs in an ovalbumin OTII model. In contrast, *Ccr7*^−/−^ mice are able to control *C. burnetii* infection following phase I WCV vaccination, although this protection may be mediated by anti-phase I LPS IgG production in *Ccr7*^−/−^ mice [47]. In murine models, innate, cell-mediated, and humoral immunity appear to contribute to vaccine-induced immunity to *C. burnetii*, and antigen-presenting cells may play an important part in the orchestration of these responses. 

### 2.2. Humans

In mice, WCV-induced protective responses are associated with increased *C. burnetii*-specific antibody titers and T-cell specific responses, but is this also evidenced in humans? Indeed, B and T cells appear to be involved in various stages of natural and vaccine-induced immune responses in humans, with long-lasting *C. burnetii*-specific T cell reactivity [54,55] and antibody titers [55,56,57] persisting following exposure to *C. burnetii* or phase I WCV. Acute Q fever patient sera has revealed an early *C. burnetii* phase II-specific humoral response followed by mounting Phase I-specific IgA, IgG, and IgM antibody responses [58]. Patients with persistent infections typically exhibit strong IgG and IgA *C. burnetii* Phase I-specific responses. In line with infection responses, human seronegative volunteers receiving WCV have exhibited phase I-specific IgM responses with a minor phase II-specific IgM and IgG components, while seropositive volunteers receiving WCV displayed Phase I-specific IgA, IgG, and IgM responses along with phase II-specific IgG and complement fixing responses, mirroring the humoral profile of those experiencing a positive skin test [58,59]. Serum from a sensitized individual experiencing a positive skin test was fractionated and assayed in a mouse seroprotection test, indicating that most protective activity was associated with phase I-specific IgM fractions. Accordingly, this fraction was able to suppress *C. burnetii* growth in vivo.

In a south Australian clinical trial with more than 1600 abattoir workers, vaccination with phase I WCV resulted in seroconversion of 30–60% of vaccines [60]. Despite this limited percentage of seroconversion, there were no cases of Q fever in vaccinated individuals who had sufficient time to develop immunity prior to infectious exposure. Like murine studies, this clinical trial indicates that antibody levels are not a direct correlate of protection from infection. Data obtained from infected individuals following natural outbreaks have provided valuable insight into the magnitude and duration of *C. burnetii*-specific humoral responses. Following the 2007–2011 Q fever outbreak in the Netherlands, *C. burnetii*-specific IgG half-life averaged 318 days, exemplifying the potential for long-lasting humoral responses in humans following natural infection [57]. In another study following patients impacted by an Australian Q fever outbreak [56], approximately 20% of previously infected individuals were seronegative 3–6 years following exposure. Importantly, the influence of chronic, persistent infections or continued environmental *C. burnetii* exposure may impact these data but does not appear to account for most persistent antibody responses. Although humoral responsiveness has been routinely described in human subjects and patients, definite correlation of antibody titer/function and protection has not been established. In fact, antibody seroconversion rates generally appear to be lower and less durable than cell-mediated markers following phase I WCV vaccination [61]. Given the sporadic nature of Q fever outbreaks and challenges involving clinical trials, establishing any associations between humoral responses and level of protection may be difficult. 

Human-derived humoral data suggest a role for immunologic components beyond antibodies in long-lived protective responses following both infection and WCV vaccination. Substantial information can be gleaned from reports of the 1981–1988 clinical trial of a phase I WCV (the immediate predecessor to Q-Vax^®^). In this trial, vaccine-induced protection was measured by the absence of Q fever cases in vaccinated subjects (although complicating factors, such as post-vaccination occupational exposure to *C. burnetii*, could not be controlled). Short-term cellular and humoral responses were quantified and appeared to correlate with vaccine-induced protection [54]. *C. burnetii*-specific antibodies were detected in 80–82% of vaccinees shortly following vaccination (from several weeks to 3 months) characterized by phase I and II IgM, followed by largely phase I IgG responses 20–60 months after vaccination, with an overall seropositivity rate of 64% among vaccinees at these later time points. Despite waning antibody titers, nearly two years following WCV vaccination, cellular immune responses were maintained in approximately 95% of vaccinated individuals five years following vaccination [54].The cell-mediated response was quantified via peripheral blood mononuclear cell (PBMC) lymphocyte stimulation index (LSI) and the 87–95% LSI positivity demonstrated by vaccinees one month after vaccination was sustained five years after vaccination, with T cells being identified as the main drivers of this response. Interestingly, these responses were reported to be cross-reactive with the Nine Mile and Priscilla strains of *C. burnetii*, which reside in divergent genomic groups, displaying the potential for heterologous protection by WCV, which has since been demonstrated in animal models [62,63]. Notably, skin test reactivity has been proposed as an indicator of post-WCV vaccination immune status [64]. Skin test reactivity appears to correlate with lymphocyte transformation assays, further indicating that cell-mediated immunity is likely a key player in WCV-induced protective responses. To further investigate cellular responses to vaccination, Izzo and Marmion investigated IFN-γ responses in WCV-vaccinated individuals at low risk for natural *C. burnetii* exposure [65]. Nearly 75% of vaccinees exhibited a positive LSI accompanied by IFN-y production following vaccination. Although IFN-γ production is associated with protective responses following vaccination, it may also contribute to prolonged disease in individuals following natural infections. Specifically, IFN-γ production has been associated with post-Q fever fatigue syndrome [66] and chronic infections [67,68]. 

### 2.3. Summary

Although correlates of protection for *C. burnetii* vaccination are yet to be fully elucidated, it appears that both innate and adaptive mediators are involved in the protective response to WCV vaccination. This is particularly evident in murine models. Overall, murine and human-derived data indicate that cellular immunity may be a promising correlate of phase I WCV-mediated protection in contrast to humoral immunity. An effective Q fever vaccine would likely need to stimulate adaptive responses, specifically inducing Th1 differentiation with an appropriate innate stimulation. The unique ability of *C. burnetii* phase I WCV to stimulate long-lasting T-cell-mediated immunity is a fascinating observation. This should be further investigated, and related research may lead to key insights for Q fever vaccine development and beyond. 

In the absence of abundant human data and previously established correlates of protection for phase I WCVs, will we be able to define correlates of protection for *C. burnetii* vaccines? Many factors have contributed to this quandary: duration of protection following vaccination has not been formally established, natural outbreaks of Q fever are not abundant, and human studies with *C. burnetii* face ethical constraints. Continued investigation in animal models and humans are needed to resolve open-ended questions, such as mechanisms of antibody-mediated immunity, T cell function, suitable adjuvants or innate stimuli, and antigenic targets of immunoprotective responses, with the goal of defining correlates of protection to aid in the design of an improved Q fever vaccine.

## 3. The Role of Bacterial Antigens in *C. burnetii* Vaccine-Induced Protection

### 3.1. Antigenic Identification

Due to the apparent importance of the adaptive immune response in vaccine-mediated protection, lymphocyte-based antigen identification and epitope mapping has been carried out in murine vaccination models. Recent work has focused on identifying both T- and B-cell-specific antigenic epitopes that may be employed in Q fever vaccine design. Immunodominant antigen identification was undertaken in a murine sublethal *C. burnetii* infection model [69], revealing numerous immunoreactive antigens in convalescent mouse sera, which appeared at different time points post-infection. Antigens derived from similar methods, cloned, and tested in a murine vaccination–challenge model did not confer protection against disease as measured by splenomegaly [70]. In humans, sera from convalescent and chronic Q fever patients were examined via protein microarray, revealing IgM- and IgG-reactive proteins unique to both disease states [71]; hundreds of seroreactive antigens were identified including surface-exposed proteins such as Com1 (CBU1910). Similar studies have revealed unique seroreactive proteins in convalescent murine and human sera [72,73,74,75]. Limited data are available from studies profiling antigenic reactivity in vaccinated animals or humans, and this is an important area of investigation in the context of rational vaccine design, as antigenic epitopes may or may not overlap in convalescent and vaccinated sera. Overall, humoral antigenicity remains an important area of investigation in *C. burnetii* research, but T cell antigens may be more important in vaccine-induced immunity. 

Accordingly, more data are available regarding T-cell antigenic identification in the context of *C. burnetii* infection and vaccination. CD4^+^ T cell peptide epitopes were identified from immunodominant *C. burnetii* proteins based on high-affinity binding capacity for MHC class II H2 I-A^b^ via bioinformatic analyses [76]. These epitopes were able to stimulate CD4^+^ IFN-γ recall responses, induce Th1 responses in vaccinated mice, and lead to slight protective responses after vaccination and challenge as measured by reduced *C. burnetii* burden in the spleen and splenomegaly. Protection upon challenge was only evident in mice vaccinated with multiple peptides, and this protective response was not as potent as that of phase I WCV. Regardless, this study demonstrated the importance of Th1 responses in vaccine-induced protective immunity and the ability of antigenic *C. burnetii* peptides to confer some degree of protection following vaccination and challenge in mice. Beyond CD4^+^ cells, using bioinformatic prediction tools, a subset of putative CD8^+^ T cell *C. burnetii* Type IV secretion system (T4SS) epitopes were identified as potential MHC-I antigens [77]. Several peptides were identified using this approach and were able to elicit CD8^+^ IFN-γ recall responses and induce some level of protection when added to a live Listeria monocytogenes vaccine vector in a murine vaccine challenge model. Recently, computational epitope identification was employed in mice expressing human MHC alleles [78]. Following subcutaneous WCV vaccination and intranasal challenge in these mice, CyTOF was utilized to investigate primary and secondary adaptive immune responses. Unsurprisingly, a general correlation of activated innate cells/lymphocytes was made with positive vaccination status. Notable findings include increased expression of circulating Ly6C^+^ T Cells (both CD4^+^ and CD8^+^CD73^+^ subpopulations) by day 10 post-vaccination (or after initial infection in lieu of vaccination), increased CD44 expression on CD4^+^ T cells by day 35 post-vaccination (possibly representing T_CM_ cells), reduced circulating innate myeloid cells, and a potential circulating immune signature (CD8^+^ T_CM_, CD4^+^ T_EM_, and mature NK cells) associated with bacterial clearance in WCV vaccinated animals. Ly6C expression and its association with T cell activation, CD8^+^ memory cells, and a CD4+ Th1 phenotype is certainly reasonable in the context of existing knowledge of murine *C. burnetii* infection and vaccine responses. A Ly6C homologue has not been identified in humans [79], but these murine data are still functionally valuable and lend insight into immune populations involved in vaccine-induced protective responses. 

T-cell-specific antigen identification has been carried out in humans as well, with recent studies revealing epitopes potentially related to adaptive immune responses to natural infection. Bioinformatically predicted T cell epitopes (HLA class II) from *C. burnetii* proteins were able to induce durable T cell IFN-γ release responses in convalescent human patients from the 2007–2010 Netherlands Q fever outbreak, nearly four years following the resolution of the outbreak and presumed *C. burnetii* exposure [80]. The same set of epitopes (both HLA class I and II) were tested against T cells from patients diagnosed with persistent Q fever revealing significant epitope reactivity overlap among these patients and the previous cohort (convalescent acute Q fever patients) with a generally stronger T-cell stimulatory response in persistently infected individuals as measured by ELISpot [81]. In both studies, HLA class II epitopes were consistently more reactive than their class I counterparts, a notable observation that may shed light on human protective responses. The similarity in HLA class II epitope reactivity between patients who had cleared disease (convalescent) and those who were not able to (persistent infection) was perhaps surprising but was also accompanied by the identification of five epitopes specific in reactivity to persistent patients alone. While these data shed light on T cell epitopes of potential diagnostic and therapeutic value, further exploration is needed to define the practical value of these findings. Long-term investigation of antigenic reactivity following infection and similar examination of individuals who are vaccinated would provide unparalleled insight into these responses. 

### 3.2. LPS as a Mediator of Protective Responses

Additional insight into the protective components of *C. burnetii* have been revealed through bacterial fractionation and extraction methods. Kazar et al. tested the protective efficacy of phase I WCV and phase I trichloroacetic acid (TCA) extract in a murine model of homologous and heterologous intraperitoneal challenge [63]. Protective efficacy was demonstrated by both vaccines, with phase I WCVs exhibiting slightly improved protection as measured by splenic bacterial burden. Together, these data indicated that proteinaceous components of phase I WCV are important components of the protective response. In contrast, in studies using recombinant or naturally isolated *C. burnetii* proteins as antigenic stimuli, these proteins alone do not appear to provide sufficient immunostimulation to confer protection comparable to phase WCV [70,79,82,83,84]. Important caveats to these data are the reduced number of individual protein antigens in historical subunit formulations and the varying influence of choice of adjuvant(s).

Phase I LPS appears to be an integral protective component of WCV-induced protective responses. *C. burnetii* undergoes LPS phase variation in vitro which results in truncation of phase I LPS to a phase II counterpart [25]. Recently, the phase variation process was directly attributed to genetic mutations in LPS biosynthesis genes [25]. Accordingly, in matched strains, phase variation does not appear to affect other *C. burnetii* components such as proteins [85]. The requirement for phase I LPS in WCV-induced protective immunity was first demonstrated by Williams et al. in an A/J mouse vaccine–challenge model [86]. Zhang et al. built on these findings using a BALB/c vaccine–challenge model [24]. By comparing the protective efficacy of phase I and II WCVs in this model, distinct protective responses were revealed. Phase I WCV induced strong protection in contrast to phase II WCV, as measured by degree of splenomegaly. Importantly, by demonstrating the protective efficacy of phase I LPS and phase II WCV plus phase I LPS that was indistinguishable from phase I WCV, the important nature of this antigen was established for protective responses. Again, the LPS purity is questionable, but these results indicate a minor role for protein antigens in the protective response. Despite extensive analysis, no exclusive correlates of protection emerged from this study. Unique protein and non-protein-based IgG-reactive compounds in vaccinated sera were identified for phase I and II WCV vaccinated mice, potentially revealing antigenic targets associated with protection. Further, although phase I and II WCV-induced immune responses were similar in makeup, the magnitude of the former was higher, possibly implicating Th1 and IgG2 responses as correlates of protection in the murine model. 

Based on the observation that phase I LPS plays a major role in WCV-induced protective responses, Peng et al. developed an LPS peptide mimic vaccine based on protective monoclonal antibody (mAb) reactivity [86]. This study demonstrated the ability of a phase I LPS-specific mAb to inhibit infection in a murine model when pre-treated with *C. burnetii* prior to infection. This suggested a direct role for phase I LPS epitopes in protective responses. Additionally, a LPS mimic peptide-keyhole limpet hemocyanin (KLH) conjugate vaccine candidate demonstrated immunogenicity and protective potential against *C. burnetii* challenge, although not to the same degree as that of phase I WCV. These data paired with human phase I-specific adaptive responses following vaccination and challenge suggest that phase I LPS (O-antigen) epitopes are major players in protective *C. burnetii* responses. 

Further investment is warranted in antigen discovery for Q fever vaccine development to aid in the rational design of modern vaccines. Additionally, a better understanding of the role of phase I LPS and relevant protective epitopes in vaccine-induced protection is needed. Although significant advancements have been made in both areas, much remains unknown, and this information is likely integral for an improved Q fever vaccine that retains potent efficacy and durability.

## 4. Current Approaches in Q Fever Vaccine Development

The future of Q fever vaccine development may be defined by the desired features of a next-generation, improved vaccine. The use of Q-Vax^®^ and other phase I WCVs as theoretical stepping stones will likely allow for the rational establishment of these features. Phase I WCVs display many advantages including potent immunogenicity, stimulation of durable immunity, and a single-dose immunization regimen. The downsides of phase I WCVs include the potential for a PVH response and the accompanying cumbersome pre-vaccination screening process along with production difficulties (e.g., high containment settings, US DSAT Select Agent status; Figure 1a). Through these observations, it is evident that an improved Q fever vaccine would mitigate the potential for a post-vaccination hypersensitivity response, induce protection at a comparable level to phase I WCVs, and be administered via a single-dose regimen. In this spirit, several approaches have been undertaken, including modification of WCVs, subunit vaccines, and LPS-based vaccines (Figure 1b). 

### 4.1. Modified WCVs

Modification of *C. burnetii* WCVs began with chemical extraction methods to mitigate reactogenicity. Chloroform–methanol extraction was used to generate chloroform–methanol residue (CMR) vaccines for Q fever [16]. Broadly, this process results in the separation of lipids (extract) and proteins (residue) [87]; multiple studies using animal models have demonstrated reduced reactogenicity and varying levels of protection of phase I CMR vaccine compared to phase I WCV [16,18,88,89,90,91,92]. Moving beyond animal models, human trials were performed with phase I CMR due to the fact of its promise as an immunogenic, non-reactive vaccine candidate in animal models. An initial human clinical study employed a single dose of CMR resulting in minimal adverse reactions at lower doses (30 µg and 60 µg), although dermal reactions were observed in several participants at 120 and 240 µg doses. Durable humoral and cell-mediated responses were noted in participants receiving 120 and 240 µg doses, with 73–90% of participants mounting *C. burnetii*-specific IgM responses, 20–40% mounting *C. burnetii* phase II-specific IgG responses, and 30–40% displaying PBMC T cell proliferation responses to *C. burnetii* [93]. Next, the CMR vaccine was tested in a prime-boost, escalating dose regimen (0.3–60 µg), demonstrating safety and immunogenic responses after the second immunization [17], and the importance of a two-dose regimen for the CMR vaccine. Recently, the CMR vaccine was administered intratracheally or subcutaneously to BALB/c mice three times followed by intratracheal *C. burnetii* challenge [94]. This study revealed enhanced lung mucosal immune responses following repeated intranasal vaccination compared to that of subcutaneous vaccination, as demonstrated by increased IgA levels in bronchoalveolar lavage fluid [94]. Additionally, general splenic T cell cytokine production was enhanced following intratracheal vaccination. Following intratracheal vaccination and subsequent challenge, animals did not display enhanced protection compared to their subcutaneous counterparts, except for reduced lung bacterial burden. The mucosal responses stimulated by this unique route of vaccination are intriguing and should be included in future Q fever vaccine development considerations. 

The development of modified WCVs has evolved along with the development of genetic tools for *C. burnetii*. Accordingly, a recent study demonstrated protective efficacy of genetically modified phase I WCV in a guinea pig vaccine–intraperitoneal challenge model [50]. By genetically removing the *C. burnetii dot/icm* locus, which encodes the bacterium’s Type IV Secretion System (T4SS), the resultant strain retained phase I LPS expression but was avirulent in a high-dose (10^6^) guinea pig intraperitoneal challenge model. As this avirulence is due to a non-revertible, large genetic deletion, these data demonstrate potential for DSAT select agent exclusion, a clear advantage for vaccine production efforts. This genetically modified WCV also demonstrated protection comparable to phase I WCV but did not completely mitigate reactogenicity, suggesting that the T4SS is dispensable for vaccine-induced protection but not solely responsible for reactogenicity. While efforts to modify WCVs capitalize on the potent protective efficacy of *C. burnetii* and its seemingly multifactorial antigen repertoire, issues with immunogenicity and reactogenicity remain. Notably, the development of an axenic media for *C. burnetiid* [36] has improved production prospects for whole-cell-based vaccines, simplifying culturing methods, and eliminating the need for potentially allergenic components such as eggs. In the future, identification of the bacterial component(s) responsible for reactogenicity will likely aid in the development of non-reactive, modified WCVs.

### 4.2. Subunit Vaccines

Subunit vaccines are typically composed of antigenic components (e.g., purified or recombinant proteins, peptides, and polysaccharides) [95] with supplemental adjuvant due to the relatively dampened immunogenicity compared to WCVs. Subunit vaccines are particularly valuable for potentially reduced reactogenicity and adverse reactions, a clear concern with existing *C. burnetii* WCVs. As previously noted, subunit vaccines have been developed for *C. burnetii* but have not demonstrated comparable protection as that of phase I WCV. Early subunit vaccine efforts were conducted by Williams et al. and involved the creation of multivalent subunit vaccines [84]. This study demonstrated the ability of purified P1 (CBU0311) to induce some degree of protective immunity following intraperitoneal challenge based on reduced splenomegaly and spleen *C. burnetii* burden compared to mock-vaccinated mice, although phase I WCV was not included in these experiments as a positive control. Subsequent efforts to identify and evaluate proteins suitable for subunit vaccine use yielded an outer membrane protein composed of 17 amino acids and devoid of LPS, which induced immunogenicity (both in humoral and cell-mediated immune assays and post-vaccination skin testing assays) and protective efficacy in rodent models [96]. The potential for contamination of isolated proteins with additional *C. burnetii* components, such as LPS, may play a role in the protective efficacy demonstrated by some isolated proteins. Beyond isolation of bacterial proteins for subunit vaccines, recombinant-based approaches have been employed. Zhang et al. demonstrated a lack of protective efficacy of individual recombinant proteins Mip (CBU0630), P1 (CBU0311), and P28 (CBU0952) in a BALB/c murine vaccine–challenge model [70]. Improved efficacy was achieved by fusing several recombinant proteins together (e.g., p1 and HspB) as opposed to using single proteins [82]; both vaccine candidates contained Freund’s adjuvant and were administered on a two-booster schedule. Although reduction of splenomegaly was comparable to that of phase I WCV-immunized mice, splenic *C. burnetii* loads appeared to be higher in P1-HspB-vaccinated animals following challenge. Despite the addition of adjuvant and booster vaccinations, this multivalent recombinant subunit vaccine was unable to completely match the protection afforded by a single administration of phase I WCV, demonstrating the powerful protective effect of this formulation. 

The apparent requirement of additional stimuli beyond proteinaceous subunit components and single adjuvants has been appreciated, as evidenced by several recent vaccine design strategies. In accordance with the importance of phase I LPS in vaccine-induced protective responses, an LPS peptide mimic-KLH conjugate vaccine was developed by Peng et al. based on mAb screening, demonstrating some protective efficacy, albeit not to the level of phase I WCV [86]. Harnessing the power of multiple adjuvants, Gilkes et al. designed and evaluated a multivalent, recombinant subunit vaccine formulated with multiple toll-like receptor (TLR) adjuvants utilizing adjuvant-linking technology to reduce the possibility of toxicity and enhance immunogenicity [83]. Incorporating previously identified putative proteinaceous *C. burnetii* antigens into this adjuvant system, immunogenicity and protective efficacy of resultant vaccines were evaluated in murine and guinea pig models. Various vaccine formulations demonstrated promising protective efficacy in a guinea pig vaccine–intratracheal challenge model following a booster vaccination and 49 day prime-challenge window. Body weight change, lung histopathology scores, and lung weight were comparable between phase I WCV and subunit vaccinated animals following challenge, and splenomegaly appeared to be consistently lower in phase I WCV vaccinated animals. The use of various adjuvant formulations yielded insight into innate immune mechanisms associated with vaccine-induced protective immunity. Specifically, TLR4, 7, and 9 appeared to be imperative for optimal subunit vaccine-induced protective responses, presumably (at least in part) due to the fact of their role in the induction of Th1 effector responses. Although these subunit vaccines require a multi-dose regimen, this demonstration of protective efficacy is promising. Post-vaccination hypersensitivity modeling revealed a range of dermal histopathologic reactions among the subunit vaccine candidates, some comparable to phase I WCV. Notably, *C. burnetii* antigens in the absence of adjuvant did not induce reactogenicity, indicating that *C. burnetii* antigens combined with the innate stimuli of adjuvants may be required for induction of the hypersensitivity response observed following phase I WCV vaccination. Subunit vaccine development for *C. burnetii* has been hindered due to the lack of understanding of immunogenic antigens and epitopes and the apparent requirement for additional stimuli beyond single proteinaceous antigens for strong protective responses. Novel antigen and adjuvant formulations will likely be introduced for *C. burnetii* subunit vaccines in the future, and these candidates will need to address the issues of sufficient immunogenicity, the implications of multi-dose immunization regimens, and reactogenicity.

### 4.3. Coxiellosis Vaccines

Vaccines for the livestock disease caused by *C. burnetii*, Coxiellosis, are a subject of current study and past developments. Coxiellosis is marked by abortion and infertility and can be rapidly spread among livestock, causing significant economic losses, livestock morbidity, and initiating human Q fever outbreaks. Accordingly, livestock vaccination has been implemented as a preventative and control measure. Inactivated WCVs have been primarily utilized for this purpose. Adverse effects, such as injection site reactions and increased body temperature, can occur in livestock following WCV vaccination [97], although the immunologic mechanisms are unknown. CMR vaccines have been proposed for *C. burnetii* control in livestock, as they have demonstrated reduced reactogenicity in both livestock and humans [98]. *C. burnetii* vaccination of livestock has yielded variable results in terms of protective efficacy and population control [6,99]. Generally, it appears that vaccination of uninfected animals may lead to protection, but the same cannot be said for that of pre-infected animals. Currently, two *C. burnetii* vaccines, Coxevac^®^ (CEVA Santé Animale, Libourne, France) and Chlamyvax FQ^®^ (Merial, Lyon, France), are available for veterinary use in many parts of the world. These vaccines are inactivated WCVs, with Chlamyvax FQ^®^ also containing inactivated *Chlamydophilia abortus*, a bacterial pathogen also known to cause adverse fertility effects. Coxevac^®^ is a phase I vaccine, while Chlamyvax FQ^®^ is a phase II vaccine. The LPS composition of these WCVs appears to affect protective efficacy as demonstrated by a goat vaccination study showing reduced bacterial shedding and abortions in phase I vaccinated animals compared to that of the phase II vaccinated animals which experienced no clinical benefit [100]. The questionable efficacy and cumbersome implementation of veterinary *C. burnetii* vaccination has contributed to limited vaccine usage in this context. Although this article focuses on Q fever vaccines for human use, the use of Coxiellosis vaccines for pathogen control at the reservoir level appears to be a promising strategy to prevent future naturally occurring outbreaks of *C. burnetii* associated with livestock that may be transmitted to humans [101,102]. Accordingly, further development of Coxiellosis vaccines would be valuable for animal and human health alike.

## 5. Looking towards the Future: Considerations for an Improved Q Fever Vaccine

*C. burnetii* is a unique pathogen due to the fact of its intracellular lifestyle, natural history, and ability to induce complex immune responses that are not yet fully understood. Currently, most studies have utilized phase I WCV to probe *C. burnetii* vaccine immune responses and identify protective bacterial components in both animal models and humans. Although phase I WCV was first introduced for Q fever prophylaxis shortly after its discovery in the 1930s, the only licensed Q fever vaccine of modern times, Q-Vax^®^, is a modern iteration of the original version. Years of progress have unfolded in the study of *C. burnetii* and Q fever, resulting in enhanced understanding of this pathogen and its interactions with host organisms. Accordingly, novel vaccine development strategies, such as modified WCVs, subunit vaccines, and LPS mimic vaccines, have been employed and evaluated in both animal models and humans. Additionally, basic scientific and clinical studies have contributed to our understanding of *C. burnetii* virulence determinants, protective responses, protective antigens, and more. 

Despite this progress, a substantial barrier in the development of improved Q fever vaccines appears to be a lack of understanding of the antigenic determinants of *C. burnetii* and protective correlates of Q fever vaccination. Potent immunity induced by phase I WCVs appears to require a complex milieu of bacterial components, likely including phase I LPS, proteins, and bacterial structures serving as adjuvants. Additionally, the stimulatory effects of phase I WCV may be related to the potential slow elimination of WCV and persistent local immune stimulation, although this remains to be investigated. Vaccine design strategies ignoring this complex interplay are likely to encounter challenges. As such, an understanding of the bacterial determinants of immunity and resultant host immune responses will be extremely valuable for continued vaccine development strategies. 

Additional considerations for modern Q fever vaccine development include the (1) importance of heterologous protection, particularly against epidemic strains; (2) route of vaccination; (3) continued development and refinement of animal models.

(1) Genomic typing efforts have revealed diverse *C. burnetii* strain composition, with eight genomic groups established by diverse genomic analysis methods [103,104,105]. In animal models of infection, these strains yield varied virulence profiles [23,63] and have been associated with acute and persistent or chronic forms of Q fever disease by genomic grouping [105]. Further, distinct phase I LPS O-antigen profiles have been associated with diverse *C. burnetii* strains [106]. Historic data suggest that LPS heterogeneity does not alter vaccine immunogenicity and efficacy [84], and monovalent WCVs appear to induce heterologous protection in animal models [63,84]. Observations in humans also support this notion, as phase I WCVs used in humans yielded protection against presumably heterogeneous challenge strains. Together, these observations support the idea that phase I WCVs offer heterologous protection and that protective antigens are shared among genetically diverse strains of *C. burnetii*, offering good news for future Q fever vaccine development efforts. Regardless, with novel *C. burnetii* strains emerging worldwide, the potential for strain-specific antigenic escape is a relevant consideration, particularly considering more reductionist vaccine approaches such as subunit formulations; 

(2) Historically, the subcutaneous route has been employed to administer *C. burnetii* phase I WCV to humans and animals. While this has not appeared to alter protective efficacy, as this vaccine appears to be capable of inducing potent immune responses, consideration of alternative routes of vaccination is intriguing for several reasons. First, the route of immunization may be important for non-WCV vaccines that are not as immunologically potent. Stimulation of mucosal immunity, as demonstrated by intratracheal administration of CMR in mice [94], may enhance local immunity to *C. burnetii*, aiding in modern vaccine design efforts. Secondly, the route of exposure may have a mitigating effect on post-vaccination hypersensitivity. Lastly, with general advancements in non-parenteral vaccine delivery [107] and renewed interest in these strategies due to the ongoing COVID-19 pandemic [108], similar approaches could be promising for Q fever vaccine development; 

(3) Animal modeling remains an integral tool in our understanding of *C. burnetii*–host interactions. Recent advancements, such as the validation of intratracheal *C. burnetii* infection models from rodents to non-human primates [109], are important for continued advancement towards an improved Q fever vaccine. The transient nature of natural Q fever outbreaks and ethical considerations limit the practicality of human clinical trials to demonstrate vaccine efficacy. As such, the US FDA’s Animal Rule will likely be utilized in the licensing of a future Q fever vaccine as demonstrated by the 2015 approval of BioThrax^®^ (Emergent BioDefense Corporation, Lansing, MI, USA) for Anthrax post-exposure prophylaxis [110]. Further validation and development of animal models of Q fever and post-vaccination hypersensitivity responses will likely be important for this purpose, among others.

The path to a non-reactive Q fever vaccine that is simple to produce and administer and that induces potent and durable protection against *C. burnetii* (Figure 1c) will be paved by continued studies aiming to decipher the complex interplay between *C. burnetii* and the host immune system. Indeed, many novel vaccine technologies spark interest but require knowledge of antigenic targets and immunologic responses that the *C. burnetii* field has not yet ascertained. As we continue to unravel the mechanisms of *C. burnetii*–host interactions we will move closer to a next-generation Q fever vaccine that will benefit people around the world.

## Figures and Tables

**Figure 1 pathogens-10-01223-f001:**
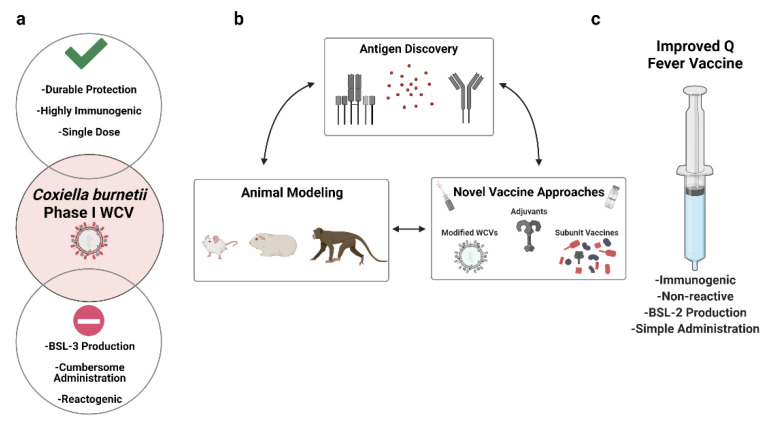
Q fever vaccine development strategies. (**a**) The currently licensed phase I whole-cell vaccine (WCV), Q-Vax^®^, is characterized by logistical pros (green check mark) and cons (red circle); (**b**) experimental tools associated with advancement towards (**c**) an improved Q fever vaccine.

## Data Availability

Not applicable.

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
