# Peer review of "Q Fever Vaccine Development: Current Strategies and Future Considerations"

_pathogens, 2021, doi:10.3390/pathogens10101223_

Round 1
Reviewer 1 Report
The article “Q Fever Vaccine Development: Current Strategies and Future Considerations” sent for consideration at MDPI Pathogens is a well-written and complete review regarding the efforts in the development of an improved Q fever vaccine.
First of all, I would like to thank the journal and the author for the opportunity to review this great text. However, some changes are suggested.
Minor corrections:
The following citations have an extra comma between the Surname of the authors and the term “et al.” that should be removed: Line 50 (Ormsbee, et al.), line 52 (Lackman, et al.), line 142 (Read, et al.), line 365 (Kazar, et al.), line 382 (Williams, et al.), line 383 (Zhang, et al.), line 398 (Peng, et al.), line 496 (Zhang, et. al.)
Lines 222-223: “(assuming these individuals had time to develop immunity following vaccination prior to natural exposure)”.
- integrate this sentence into the text by removing the parentheses.
Line 254: PBMC – please provide the full term before start using initials.
Lines 266, 268: Substitute the letter “y” with the representative gamma letter “γ”
Line 402: Please provide the full term for KLH (keyhole limpet hemocyanin)
Line 441: “120- and 24-μg doses”. By the context, I believe it should be 240- μg.
Line 466: Please provide the full term for DSAT (Division of Select Agents and Toxins)
Line 514: Please, provide the term for TLR (toll-like receptor)
Lines 553 – 554: It is emphasized the existence of two vaccines for coxiellosis, Coxivac (CEVA) and Chlamyvax FX (MERIAL).
- I believe the correct name of the vaccine is Chlamyvax FQ, not FX.
- It is also important to briefly discuss the differences between both vaccines in terms of compositions (Coxevac – phase I; Chlamyvax FQ – phase II) and protection.
Suggested literature:
-
- https://pubmed.ncbi.nlm.nih.gov/16005747/
- Chapter 24 - Q fever vaccination in ruminants: a critical review. In The Principles and Practice of Q Fever.
- https://www.clinicalmicrobiologyandinfection.com/article/S1198-743X(14)63539-4/fulltext
Reviewer 2 Report
Manuscript: pathogens-1373667
"Q Fever Vaccine Development: Current Strategies and Future Considerations" by Carrie Mae Long
The Review by Carrie Mae Long discussed the history, current approaches, and future considerations of human Q fever vaccine development.
Points of critique:
- In parts of the review (especially at the beginning of Chapter 1), text sections are missing appropriate references. In addition, uniform spelling is not always consistent in the text. Further correction is recommended.
- It is recommended that the author make it clear at the beginning of the review that it focuses primarily on human vaccine development.
- The discussion of the M-44 vaccine should be more differentiated. Several research groups (Genig et al. 1965, Sterkhova 1965, Zdrodovskii & Genig 1962, and Johnson et al. 1976) described that this Ph II live vaccine induces immunization. Robinson and Hasty (1974) further reduced the virulence properties of M-44 and showed that the vaccine provided protection against both Ph II and Ph I bacteria. It is recommended that the author should adjust her subjective assumption ("... presumably due to low protective efficacy.").
- It would be desirable for the author to address more clearly the fact that Ph II WCVs also exhibit protection, albeit reduced, and to appreciate the fact that LPS per se, regardless of phase variation, is critical for vaccine development.
